# Multi-Dimensional Game-Theoretic capability evaluation under environmental shifts

## Abstract

Driven by advances in deep learning, intelligent game-theoretic strategies have progressed rapidly. As these methods enter pivotal real-world domains, rigorous and impartial evaluation has emerged as a central concern. Although many benchmarks now ensure the comparability of diverse gaming algorithms, prevailing evaluation methods focus on macro-level win–loss outcomes and aggregate metrics, providing little insight into differences among strategies across dimensions of game capabilities. In addition, existing methods are limited to evaluation within a single environment and struggle to handle evaluation tasks under environment shift. In this work, we propose MGCE (Multi-dimensional Game-theoretic Capability Evaluation), a novel evaluation framework that first partitions the scenarios and embeds them using large-model-guided multi-modal features. It then leverages neural networks to model strategy–environment interactions, quantifying multi-dimensional capabilities and characteristics of the environment. MGCE further incorporates a fine-tuning module to predict the performance of game strategies under shifting environments. The experimental results demonstrate that the multi-dimensional game capabilities learned by MGCE facilitate interpretable analysis of strategies, and that MGCE outperforms competing baselines in performance prediction across both private and public environments.

## 1 Introduction

The focus of artificial intelligence research is gradually shifting from the paradigm of perceptual intelligence to that of decision-making intelligence. Within this transition, game theory provides decision makers with a mathematical framework to characterize strategic interactions among agents and their possible outcomes. In recent years, empowered by the representational power of neural networks, game theory and deep learning have converged and demonstrated game-theoretic decision-making prowess surpassing that of humans in fields such as Go Silver et al. (2016; 2018), video games Vinyals et al. (2019); Huang et al. (2023), autonomous driving Liu et al. (2022; 2023a), and robotic control Pinto et al. (2017); Haarnoja et al. (2024). Given the importance of game theoretic decision making, its extension to broader real-world applications has attracted widespread attention. Nevertheless, high modeling complexity, prohibitive sample and computational costs, and limited generalization continue to impede practical deployment.

Actually, it is crucial to establish a comparability and reproducible framework for the evaluation of game-theoretic decision-making methods before pursuing real-world applications. A wide array of benchmarks for evaluating game strategies has emerged in recent years Mordatch & Abbeel (2018); Lanctot et al. (2019); Bard et al. (2020). By testing average returns or win rates against opponents, different strategies can be roughly compared. Building on this foundation, evaluation methods such as Elo Elo (1978) and $\alpha$-Rank Omidshafiei et al. (2019) derive rating scores from win–loss relationships, thereby generating an ordered ranking across multiple strategies. However, the above methods crudely compress complex game capabilities into a single overall score, lacking insight into the characteristics of different aspects of strategy. Thus, these methods are unable to provide a thorough analysis of the underlying causes when a strategy fails. Moreover, their perspective remains narrowly confined to the strategies, overlooking the role and dynamics of the game environment. In other words, the test environment may present cross-domain challenges, including maps, rules, and even opponent strategies that differ from those in training, up to a full transfer from simulation to the real world. Although several studies have introduced unified performance metrics

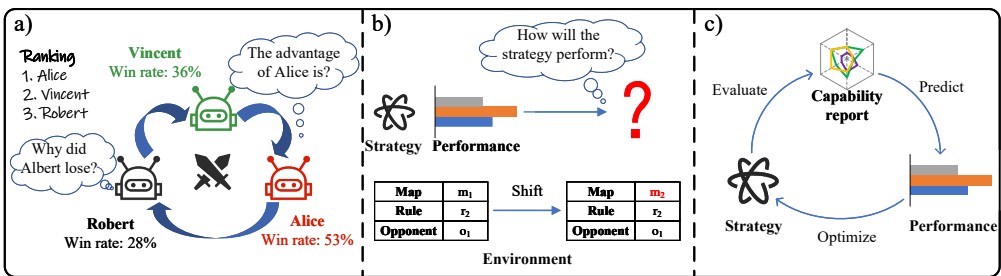

Figure 1: A concise overview of the problem. a) Conventional methods collapse evaluation into a single aggregate metric, obscuring the advantages of the algorithm and the reasons for failure. b) Existing methods are limited to single environments and struggle to handle cross-environment evaluation tasks. c) We generate multi-dimensional capability reports of game strategies, and further predict their performance under changing environments. This further supports the refinement of strategies.

across heterogeneous environments Jordan et al. (2020), there remains a notable lack of quantitative analysis of how the performance of strategies varies under specific environmental shifts.

Unlike previous strategy-evaluation methods, we do not limit the analysis to a single performance metric, which would overlook the multi-dimensional nature of strategic capability in games. Actually, different game strategies may excel or falter across distinct dimensions of capability, yet such fine-grained differences are often obscured by macro-level metrics, such as win rate. In addition, the inherent difficulty of the game environment and its sensitivity in distinguishing different dimensions of capability can significantly shape the objectivity of strategy evaluation. Inspired by cognitive diagnosis, a research field in educational psychology, which analyzes the responses of examinees to test items and infers their mastery of different knowledge concepts. Similarly, game strategies can be regarded as examinees, while game environments can be viewed as test items. By evaluating how strategies perform across diverse environments, one can quantify the various dimensions of capability a strategy exhibits in a task, as well as capture characteristics such as the difficulty and discriminative power of the environments. Furthermore, we extend our evaluation to the generalizability of game strategies, aiming to predict their performance across environments. This not only reflects how well a strategy adapts to novel settings, but also reduces the human and material costs of repeated testing, thereby enhancing the efficiency of strategy optimization.

In this paper, we propose an evaluation framework for game strategies, MGCE, designed to assess the multi-dimensional capabilities of strategies and to predict their performance under environmental shifts. As illustrated in Figure 2. MGCE first partitions the gaming task into distinct scenarios and then extracts multimodal feature vectors to represent these scenarios. It then introduces an evaluation module, which leverages performance data from game strategies to assess their fine-grained capabilities, while simultaneously deriving the characteristics of the test scenarios. It further employs a cross-environment prediction approach to infer how game strategies would perform in novel environments. The main contributions of our work can be summarized as follows:

- Unlike existing evaluation methods that assign strategies a single overall score, we introduce the concept of cognitive diagnosis to evaluate strategies in greater detail.

- We propose a novel evaluation framework, MGCE, that is capable of quantifying the multi-dimensional capabilities of game strategies while predicting their performance under environment shift.

- Empirical results demonstrate that the fine-grained game capabilities obtained by MGCE enable interpretable analyses of strategy strengths and weaknesses. Furthermore, across two settings, MGCE surpasses existing approaches in cross-environment prediction of game performance.

## 2 RELATED WORK

**Game-theoretic benchmark platforms.** Empirical research on game-theoretic algorithms benefits from an increasingly rich suite of benchmarks. The Arcade Learning Environment Bellemare et al. (2013) unifies interfaces and evaluation protocols across dozens of Atari 2600 games, catalyzing progress in single-agent reinforcement learning. Complex adversarial environments exemplified by StarCraft Multi-Agent Challenge (SMAC) Samvelyan et al. (2019) and Google Research Football Kurach et al. (2020) provide controlled, reproducible proving grounds for team coordination, fine grained micromanagement, and long horizon control. Oriented toward social interaction and mixed-motive, Overcooked-AI Carroll et al. (2019), Hanabi Bard et al. (2020), and Multi-agent Particle Environment (MPE) Mordatch & Abbeel (2018) emphasize collaboration with unfamiliar partners, intent inference, and communicative competence. OpenSpiel Lanctot et al. (2019) further introduces a general, unified platform that integrates multi-agent environments and algorithms, spans a broad spectrum of game tasks, and enables performance comparisons across paradigms. Turning to more real-world applications, PGSim Chen et al. (2021) provides an efficient, high-fidelity power grid simulation platform for training and evaluating coordinated grid control policies. Overall, existing benchmarks provide robust support for evaluating game strategies from different perspectives.

**Game-theoretic strategy evaluation methods.** Despite the improved comparability afforded by diversified benchmarks, it remains common to use coarse-grained metrics such as success rate, win-loss record, and average return. To remedy this limitation, a growing number of scholars have proposed more evaluation methodologies. Elo Elo (1978) and TrueSkill Herbrich et al. (2006) rating systems infer overall strength from match outcomes, yet they remain sensitive to the distribution of opponents. To capture the surrounding strategic ecology, $\alpha$-Rank Omidshafiei et al. (2019) and Nash averaging Balduzzi et al. (2018) conduct evolutionary or equilibrium analyses at the strategic metagame level to reveal population dynamics and structural advantages. However, existing evaluation methods remain confined to macro-level composite scores, offering little insight into how strategies differ across capability dimensions. Furthermore, they focus on the performance of game strategies within fixed environments, neglecting the analysis under specific environment shifts. We therefore introduce an evaluation paradigm grounded in multi-dimensional capability diagnosis, with the capacity to predict strategy performance across environments.

**Cognitive diagnosis research.** Cognitive diagnosis originated in educational psychology. The canonical exemplars are DINA De La Torre (2009); Von Davier (2014) and IRT Embretson & Reise (2013), which cast response outcomes as interactions between respondent traits and item characteristics, yielding probabilistic estimates of correctness. Multi-dimensional IRT Reckase (2009) generalizes scalar parameters to vectors, enabling simultaneous modeling of multiple latent traits and their correlations. In addition, matrix factorization methods such as SVD Toscher & Jahrer (2010) and PMF Mnih & Salakhutdinov (2007) have been used to learn latent trait vectors for respondents and items. Recently, cognitive diagnosis has deepened its integration with neural networks. NeuralCDM Wang et al. (2020; 2022) project students and exercises to factor vectors and incorporates neural networks to learn the complex exercising interactions. Drawing on these insights, we introduce the principles of cognitive diagnosis into the evaluation of game-theoretic algorithms. Specifically, we cast strategy-environment interactions as student-exercise responses, represent environment characteristics with multi-modal data, and employ neural networks to capture latent traits that arise in strategic play, jointly quantifying strategy capabilities and environment characteristics. Furthermore, with limited feedback, we predict performance changes under environmental shift, enabling evaluation through capability profiles and cross-environment transferability.

## 3 THE PROPOSED ALGORITHM

### 3.1 PROBLEM DEFINITION

In this section, we present the preliminary knowledge required for our study and formally define the problem addressed in this work.

Given a game task $\mathcal{T}$, the task environment is initially partitioned into a collection of scenarios $S = \{s_1, s_2, \cdots, s_m\}$, according to the specific $d$ capabilities under consideration. For each trained strategy $A = \{a_1, a_2, \cdots, a_t\}$, conduct $n$ independent runs within every test scenario. The average

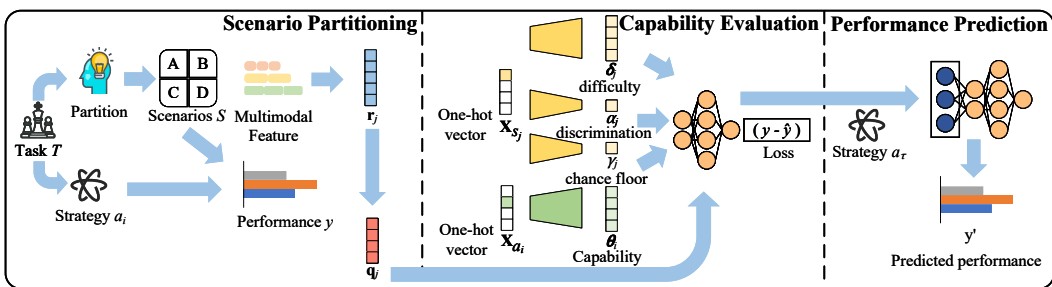

Figure 2: Conceptual framework of the proposed MGCE.

performance of each strategy in a given scenario is then recorded as its test result. In this work, our task is to quantify the multi-dimensional capabilities of game strategies based on performance data.

As shown in Figure 2, MGCE first represents each scenario using multimodal data to produce representation vector $\mathbf{r}_j$, then learns a capability-related vector $\mathbf{q}_j$. It takes one-hot encodings $\mathbf{a}_i$, $\mathbf{x}_{s_j}$ of the strategy and the scenario and models their interaction with a neural network. Through loss optimization, MGCE infers latent strategy capabilities $\boldsymbol{\theta}_i$ and scenario characteristics $\boldsymbol{\delta}_j$, $\alpha_j$, $\gamma_j$ from performance data. It further fine-tunes parameters with an anchor strategy $a_\tau$, and outputs performance prediction across environments.

## 3.2 SCENARIO PARTITIONING

Evaluating the decision-making capabilities of game algorithms requires not only quantifying their effectiveness and distinguishing their relative strengths and weaknesses, but also providing a solid foundation for deployment in real-world conditions. Existing studies primarily rely on the win–loss outcomes of game algorithms as evaluation metrics, and numerous benchmarks have already been established for testing purposes. However, these approaches overlook the fact that gaming is a complex decision-making process, in which differences in strategic capabilities across multiple dimensions can directly influence outcomes. When evaluation is reduced solely to win–loss results, it becomes difficult to trace the underlying causes of success or failure, thereby limiting explainability.

Therefore, we partition the game task into several scenarios according to different capabilities (such as cooperation and deception), with each scenario corresponding to the examination of a particular capability. Furthermore, we introduce random perturbations (such as initial coordinates and available resources) into each scenario to generate a set of scenarios that serve as the testing environment. The prior association between scenarios and capabilities is represented by a binary mapping matrix $\mathbf{Q} \in \mathbb{R}^{m \times d}$, where an entry $Q_{i,j} = 1$ indicates that scenario $s_i$ involves $j$-th capability. For each scenario, multi-modal features $\mathbf{e}_i$ are utilized to construct its embedding. For example, using convolutional neural networks Long et al. (2015) to extract image features. Alternatively, leveraging BERT Devlin et al. (2019) to extract textual features that capture the underlying rules and constraints. In this work, we further utilize the powerful comprehension abilities of multi-modal large models. Specifically, the model is tasked with interpreting the scenario data to capture the current situation and output the corresponding situational features. Then, multi-modal features are fused, with the formal representation shown below:

$$\mathbf{r}_j = \Phi([\mathbf{e}_1, \mathbf{e}_2, \cdots]). \tag{1}$$

Here $\Phi$ is implemented using a transformer model combined with average pooling.

Although the correspondence between scenarios and capabilities has been established based on expert knowledge, in practice there may still exist minor labeling deviations in the capability categories. In addition, a single scenario may simultaneously involve multiple capability dimensions. Therefore, it is necessary to further learn the capability vector representation of each scenario. First, based on the prior mapping matrix $\mathbf{Q}$, the prior capability association vector $\mathbf{u}_j \in \mathbb{R}^d$ of each scenario is obtained as follows:

$$\mathbf{u}_j = (\mathbf{Q}_{j,.})^\top, \tag{2}$$

where $\mathbf{Q}_{j,\cdot}$ denotes $j$-th row of matrix $\mathbf{Q}$. In addition, we obtain capability-related vector $\mathbf{v}_j \in \mathbb{R}^d$ based on scenario representation $\mathbf{r}_j$:

$$\mathbf{v}_j = \text{MLP}(\mathbf{r}_j), \tag{3}$$

where MLP denotes multilayer perceptron.

To obtain the final capability association vector $\mathbf{q}_j$, the representations $\mathbf{u}_j$ and $\mathbf{v}_j$ are further fused and normalized through a Softmax layer, formally expressed as follows:

$$\mathbf{q}_j = \text{Softmax}(\mathbf{u}_j + \mathbf{v}_j). \tag{4}$$

Here, the capability-related vector $\mathbf{q}_j$ is analogous to that of knowledge concepts associated with test items in educational psychology. It will serve as a structural prior for the subsequent capability evaluation module, thereby enhancing estimation stability.

## 3.3 CAPABILITY EVALUATION

Prevailing evaluations take a macro, single-dimensional view, preventing attribution of game-strategy performance differences to specific capability dimensions. Inspired by cognitive diagnosis, we leverage the test performance of game strategies to assess their fine-grained capabilities. Given the one-hot vector $\mathbf{x}_{a_i}$ of a game strategy, its capability vector is computed as follows:

$$\boldsymbol{\theta}_i = \sigma(\mathbf{W}_\theta \mathbf{x}_{a_i}), \tag{5}$$

where $\sigma(\cdot)$ is an activation function, and each component of $\boldsymbol{\theta}_i \in \mathbb{R}^d$ encodes the quantified value of the corresponding capability. Furthermore, unlike most evaluation methods, we also incorporate an evaluation of scenario characteristics. Similarly, given the one-hot vector $\mathbf{x}_{s_j}$ of a scenario, we obtain:

$$\begin{aligned}
\boldsymbol{\delta}_j &= \sigma(\mathbf{W}_\delta \mathbf{x}_{s_j}), \\
\alpha_j &= \sigma(\mathbf{W}_\alpha \mathbf{x}_{s_j}), \\
\gamma_j &= \sigma(\mathbf{W}_\gamma \mathbf{x}_{s_j}),
\end{aligned} \tag{6}$$

where difficulty $\boldsymbol{\delta}_j$ denotes the capability threshold required to attain a high success rate; the discrimination $\alpha_j$ quantifies how sharply this scenario distinguishes between strategies near that threshold; and the chance floor $\gamma_j$ is the success probability of a random strategy in this scenario. Since capability, discrimination, and chance floor are typically nonnegative, whereas difficulty may take negative values, we set $\sigma$ as Sigmoid for the former and Tanh for the latter. We then employ the three-parameter logistic (3PL) model, which jointly accounts for strategy and environmental characteristics to predict performance. Following Wang et al. (2020), who employed neural networks to fit the 2PL formulation, we extend this approach to 3PL formulation, computed as follows:

$$\hat{y}_{i,j} = \gamma_j + (1 - \gamma_j) \cdot f(\alpha \mathbf{q} \circ (\boldsymbol{\theta}_i - \boldsymbol{\delta}_j)), \tag{7}$$

where $\circ$ denotes element-wise product. Notably, the performance $y$ is not limited to win rate. It may also encompass other outcomes, such as time steps and resource consumption. To fit the performance data of game strategies in the test scenarios, we adopt the mean squared error between predicted and true outcomes as the loss function as follows:

$$\mathcal{L} = \frac{1}{t+m} \sum_{i=1}^{t} \sum_{j=1}^{m} (y_{i,j} - \hat{y}_{i,j})^2. \tag{8}$$

By iteratively optimizing this function, we achieve the quantification and updating of strategy capabilities and scenario characteristics.

## 3.4 PERFORMANCE PREDICTION

To capture latent transfer signals under environmental shifts and predict the performance of game-theoretic policies in novel environments, we introduce a fine-tuning strategy that updates the parameters of pretrained models. We first define anchor strategy for trial runs in the new environment, with the results serving as input for model fine-tuning. To furnish a stable reference across diverse

---

**Algorithm 1** Proposed algorithm

---

**Input**: One-hot encodings of game strategy $\{\mathbf{x}_{a_1}, \mathbf{x}_{a_2}, \cdots\}$ and scenario $\{\mathbf{x}_{s_1}, \mathbf{x}_{s_2}, \cdots\}$, test results $\{y_{1,1}, y_{1,2}, \cdots\}$

**Output**: Capability vectors $\{\boldsymbol{\theta}_1, \boldsymbol{\theta}_2, \cdots\}$, predicted cross-environment performance $\{\hat{y}_{1,1'}, \hat{y}_{1,2'}, \cdots\}$

 1: Partitioning scenarios and generating test scenario set $S$
 2: **for** $j = 1, 2, \cdots, m$ **do**
 3:     Calculate scenario representation $\mathbf{r}_j$ via Eq. 1
 4:     Calculate capability association vector $\mathbf{q}_j$ via Eq. 4
 5: **end for**
 6: **for** $k = 1, 2, \cdots, K$ **do**
 7:     **for** $i = 1, 2, \cdots, t$ **do**
 8:         **for** $j = 1, 2, \cdots, m$ **do**
 9:             Calculate the capability vector $\boldsymbol{\theta}_i$ of $i$-th game strategy via Eq. 5
10:             Calculate difficulty $\boldsymbol{\delta}_j$, discrimination $\alpha_j$ and chance floor $\gamma_j$ f $j$-th scenario via Eq. 6
11:             Calculate estimated performance $\hat{y}_{i,j}$ via Eq. 7
12:         **end for**
13:     **end for**
14:     Calculate loss function $\mathcal{L}$ via Eq. 8 and employ Adam optimizer to minimize $\mathcal{L}$
15: **end for**
16: Select achor strategy $a_\tau$ via Eq. 10
17: Calculate the shift performance via Eq. 11 and fine-tuning the model
18: **for** $i = 1, 2, \cdots, \tau - 1, \tau + 1, \cdots, t$ **do**
19:     **for** $j' = 1, 2, \cdots, m$ **do**
20:         Predict the performance $\hat{y}_{i,j'}$ of strategy $a_i$ in scenario $s_{j'}$
21:     **end for**
22: **end for**
23: **return** $\{\boldsymbol{\theta}_1, \boldsymbol{\theta}_2, \cdots\}$ and $\{\hat{y}_{1,1'}, \hat{y}_{1,2'}, \cdots\}$

---

scenarios and serve as a benchmark for model calibration, it must be free of conspicuous weaknesses across all dimensions of capability. Accordingly, we define the balance score as follows:

$$\Gamma(a_i) = (\prod_{k=1}^{d} \boldsymbol{\theta}_{i,k})^{1/d}. \tag{9}$$

The top-scoring strategy is then designated the anchor strategy:

$$a_\tau = \arg\max_{a_i} \Gamma(a_i). \tag{10}$$

To learn the performance drift induced by environmental shifts, we freeze the parameters of $f(\cdot)$ in Eq. 7 to $f_\Theta(\cdot)$ and prepend a calibration layer as follows:

$$\hat{y}_{\tau,j'} = \gamma_{j'} + (1 - \gamma_{j'}) \cdot f_\Theta \cdot g(\alpha \mathbf{q}' \circ (\boldsymbol{\theta}_\tau - \boldsymbol{\delta}_{j'})). \tag{11}$$

We then fine-tune using the same optimization scheme as in Eq. 8, thereby enabling inference of the performance of remaining strategies in new environments. Thus, we furnish a generalization evaluation of strategies without individually testing every new environment, thereby offering low-cost decision support for the optimization of game-theoretic algorithms and their deployment in novel settings.

The overall process of MGCE is described in Algorithm 1. Particularly, in line 1, MGCE partitions the game task into capability-related scenarios based on prior knowledge and introduces random perturbations to yield a collection of testing scenarios. In lines 2–5, MGCE derives feature embeddings for the multi-modal inputs of scenario using the corresponding embedding algorithms, then fuses these modalities through a transformer layer to obtain scenario representations. Leveraging the prior mapping matrix Q between scenarios and capabilities, it computes vector u and trains an MLP

to learn vector v. It applies softmax to obtain the final capability relevance vector for scenario j. In lines 6-15, MGCE builds a differentiable nonlinear mapping that takes one-hot encodings of game strategies and scenarios as inputs, computes a multi-dimensional game capability vector, and simultaneously estimates scenario difficulty, discrimination, and chance floor. On this basis, it utilizes Eq. 7 to compute the performance prediction and iteratively optimizes the loss with Adam optimization algorithm, thereby yielding evaluation results. In line 16, it derives the anchor strategy by computing a balanced score, and then in line 17, it fine-tunes the model via a threshold calibration layer, using the test results of the anchor strategy in new environments. In lines 18-22, MGCE predicts the performance of game strategies under environmental shifts as the final output.

## 4 EXPERIMENTS

In this section, we report our experimental settings and results in detail.

### 4.1 EXPERIMENTAL SETTINGS

We evaluate the proposed method across two environments. We first constructed a 3v1 pursuit-evasion environment as a test platform, with the goal of capturing the evader in a three-dimensional arena. We vary initial positions and constraints to evaluate hunting, efficiency, collaboration, and robustness. We benchmark single-agent algorithms SAC Haarnoja et al. (2018), PPO Schulman et al. (2017), TD3 Fujimoto et al. (2018), multi-agent algorithms including MASAC Pu et al. (2021), MAPPO Yu et al. (2022), and rule-based approaches including Voronoi-based strategy and naive pursuit strategy. We modify the game environment by introducing obstacles and varying the evader strategy.

To further substantiate that our proposed framework can interface with public benchmarks and reuse existing experimental configurations for evaluation, we test game algorithms on Multi-agent Particle Environment (MPE) Mordatch & Abbeel (2018). To constrain the motion of agents, we introduce additional walls. Within this environment, we select *simple_spread*, *simple_listener_speaker*, and *simple_tag* to evaluate capabilities across different dimensions. We adopt six discrete methods and seven continuous ones as test algorithms, with further details in the appendix. We alter the environment by introducing obstacles, which serve as the subject of cross-environmental prediction.

For every scenario across the two environments, we run 20 and 1000 times respectively, and report the mean normalized time steps, forming the **Pursuit** and **MPE** datasets. We embed terrain images with Swin-TransformerLiu et al. (2021), encode the rules as text embeddings with BERTDevlin et al. (2019). As shown in Figure 3, we leverage GPT-5 multimodal reasoning to derive situational embeddings.

We compare our MGCE with two matrix factorization (MF) methods, including SVD Toscher & Jahrer (2010), PMF Mnih & Salakhutdinov (2007), two IRT series methods, including IRT Embretson & Reise (2013), MIRT Reccase (2009), and two cognitive diagnosis methods, including DINA De La Torre (2009); Von Davier (2014), NeuralCDM(NCDM) Wang et al. (2020; 2022). Full details of the experimental setup are provided in the Appendix.

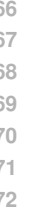
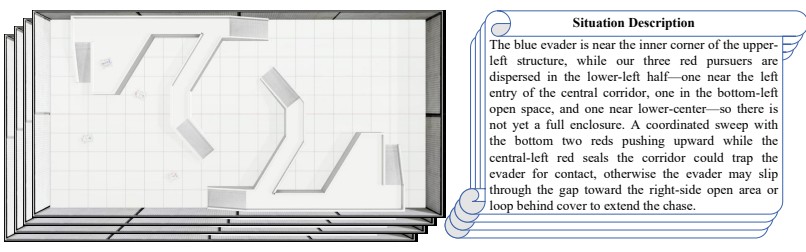

**Situation Description**

The blue evader is near the inner corner of the upper-left structure, while our three red pursuers are dispersed in the lower-left half—one near the left entry of the central corridor, one in the bottom-left open space, and one near lower-center—so there is not yet a full enclosure. A coordinated sweep with the bottom two reds pushing upward while the central-left red seals the corridor could trap the evader for contact, otherwise the evader may slip through the gap toward the right-side open area or loop behind cover to extend the chase.

Figure 3: Situation embeddings derived by multimodal large model.

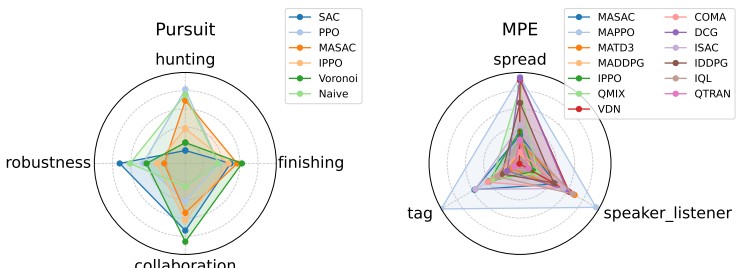

Figure 4: Radar chart of multi-dimensional game-theoretic capabilities

We evaluate the predictions using three regression metrics: MAE (Mean Absolute Error): the average absolute difference between predicted and true values; RMSE (Root Mean Squared Error): the square root of the mean of squared differences, which penalizes larger errors more heavily; and $R^2$ (coefficient of determination): the proportion of variance in the response explained by the model. In the following figures and tables, ↑(↓) means the higher (lower) the value, the better the performance is.

### 4.2 RESULT ANALYSIS

#### 4.2.1 RESULTS OF CAPABILITY EVALUATION

We first present the capability evaluation results in Figure 4. In the Pursuit environment, Voronoi-based method partitions the field and executes multi-point encirclement, achieving the best results on collaboration and finishing. PPO and SAC excel in hunting and robustness respectively, yielding overall performance that surpasses other reinforcement learning strategies. In the MPE environment, MAPPO performs strongly across all three metrics, consistent with the highest success rate observed in evaluation. These findings suggest that quantifying game theoretic strategies across capability dimensions helps expose strengths and reasons of success, and enables rapid localization of weaknesses to guide re-training.

#### 4.2.2 RESULTS OF PERFORMANCE PREDICTION

We then compare the predictive performance of our MGCE and baselines on the Pursuit and MPE datasets. We take each game strategy in turn as the anchor and report the summed metrics, and MGCE-A indicates that our proposed method employs anchor selection. In addition, we regard the success probability produced by IRT series methods as a predictor of performance. As shown in Table 1 and Figure 5, MGCE outperforms its rivals across most metrics.

MF methods possess the capability to capture latent interactions among entities and predict interaction scores. On the MPE dataset, PMF shows some predictive ability. However, its RMSE indicates that it produces predictions with significant deviations from true performance. By contrast, SVD performs better on predicting performance transfer. It uncovers latent low-dimensional relations between strategies and scenarios from anchor performance data, achieving lower prediction error. Since the IRT series handles only binary responses, continuous values are binarized with a threshold of 0.5. This entails information loss and diminishes the predictive power of these traditional methods. Compared with the IRT series, cognitive diagnostic methods exploit the mapping matrix between scenarios and capabilities. However, Figure 5 shows that DINA can hardly learn effective strategy–scenario relations. NeuralCDM leverages the capabilities of neural networks to better fit the nonlinear relationships of the response function. Compared with these methods, our proposed method better captures the mapping from environmental shift to game performance and achieves superior results. Its improvement over NeuralCDM may be attributed to the integration of scenario-rich multimodal information and reasonable fine-tuning approach. Additionally, Figure 5 also demonstrates that MGCE and MGCE-A better fit the distribution of actual performance.

Table 1: Results on Pursuit and MPE dataset. The best result is highlighted in **bold**, while the second-best result is underlined.

| Shift | | SVD | PMF | IRT | MIRT | DINA | NCDM | MGCE | MGCE-A |
|---|---|---|---|---|---|---|---|---|---|
| | | | | | Pursuit dataset | | | | |
| Terrain | MAE↓ | 0.100 | 0.163 | 0.204 | 0.373 | 0.182 | 0.119 | 0.080 | **0.077** |
| | RMSE↓ | 0.120 | 0.225 | 0.248 | 0.448 | 0.247 | 0.142 | 0.104 | **0.102** |
| | $R^2$ ↑ | 0.577 | -0.488 | -0.801 | -4.449 | -0.784 | 0.412 | 0.683 | **0.692** |
| Policy | MAE↓ | 0.129 | 0.155 | 0.220 | 0.443 | 0.166 | 0.124 | 0.104 | **0.101** |
| | RMSE↓ | 0.155 | 0.233 | 0.267 | 0.523 | 0.248 | 0.151 | **0.137** | **0.137** |
| | $R^2$ ↑ | 0.412 | -0.329 | -0.749 | -4.973 | -0.500 | 0.444 | **0.544** | 0.534 |
| | | | | | MPE dataset | | | | |
| Terrain | MAE↓ | 0.129 | 0.176 | 0.179 | 0.305 | 0.236 | 0.161 | 0.114 | **0.106** |
| | RMSE↓ | **0.170** | 0.281 | 0.228 | 0.358 | 0.410 | 0.239 | 0.201 | 0.174 |
| | $R^2$ ↑ | **0.775** | 0.384 | 0.594 | -2.259 | -0.309 | 0.556 | 0.686 | 0.756 |

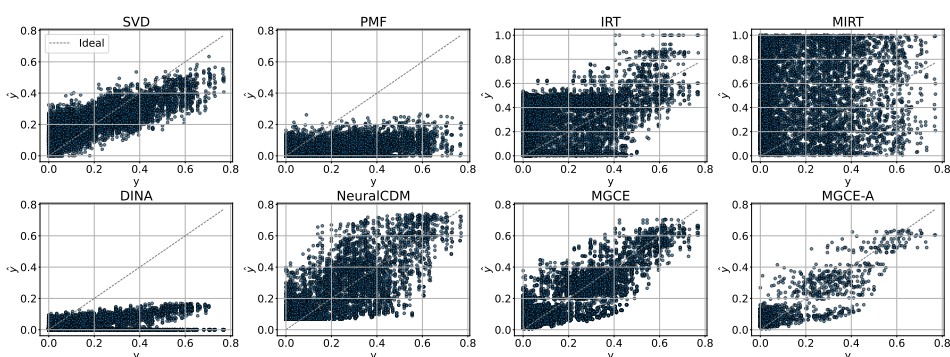

Figure 5: Predicted vs. ground-truth scatter plots under terrain shift within the pursuit-evasion dataset

Table 2: Ablation results for the scenario multimodal representation module within MGCE.

| | w/o T | w/o R | w/o S | w/o TR | w/o TS | w/o RS | w/o TRS | MGCE |
|---|---|---|---|---|---|---|---|---|
| MAE↓ | 0.115 | 0.114 | 0.117 | 0.116 | 0.121 | 0.121 | 0.118 | 0.114 |
| RMSE↓ | 0.204 | 0.206 | 0.210 | 0.207 | 0.213 | 0.212 | 0.210 | 0.201 |
| $R^2$ ↑ | 0.676 | 0.670 | 0.657 | 0.668 | 0.645 | 0.651 | 0.657 | 0.686 |

## 4.3 ABLATION STUDY

We conduct ablation study on MPE dataset to investigate the contribution of the multimodal representation in the scenario partitioning module. We use w/o T, w/o R, and w/o S to indicate ablations that remove terrain image representation, rule text representation, and situational representation, respectively. As shown in Table 2, MGCE attains the best performance, validating the effectiveness of multimodal representations. Moreover, w/o S lags behind w/o T and w/o R, underscoring the importance of situational representation.

## 5 CONCLUSION

In this paper, we study the multi-dimensional evaluation of game strategies, an important but under-explored problem. We introduce MGCE, an evaluation framework that leverages neural networks to model interactions between game strategies and environments, assess strategic capabilities, and predict performance across environments. Experimental results validate the effectiveness of MGCE.

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

## A APPENDIX

### A.1 THE USE OF LARGE LANGUAGE MODELS

In this paper, we use large language models to polish writing.

#### A.1.1 EXPERIMENTAL SETTINGS

We adopte six discrete methods IQL Matignon et al. (2012), VDN Sunehag et al. (2017), QMIX Rashid et al. (2020), DCG Böhmer et al. (2020), QTRAN Son et al. (2019), COMA Foerster et al. (2018), and seven continuous ones MASAC Pu et al. (2021), ISAC, MAPPO Yu et al. (2022), IPPO, MATD3 Ackermann et al. (2019), MADDPG Lowe et al. (2017), IDDPG as test algorithms in MPE environment. All code was implemented by Xuance.Liu et al. (2023b). All experiments are run on a Linux computer with a 12th Gen Intel(R) Core(TM) i9-12900K CPU and a 4090 GPU.

