# OpenReview forum: "Multi-dimensional game-theoretic capability evaluation under environmental shifts"
_ICLR.cc/2026/Conference — ICLR 2026 Conference Withdrawn Submission_

### Official Review · Reviewer_4Z3h · 2025-10-29

**Soundness:** 3
**Presentation:** 3
**Contribution:** 2
**Rating:** 2
**Confidence:** 3

**Summary:**

This paper proposes MGCE (Multi-dimensional Game-theoretic Capability Evaluation), a framework that evaluates and predicts the performance of game-theoretic strategies under environmental shifts. Drawing inspiration from cognitive diagnosis theory, MGCE decomposes complex game tasks into capability-related scenarios, embeds them with multimodal features, and uses neural networks to model interactions between strategies and environments. It quantifies multi-dimensional strategic capabilities and predicts cross-environment performance through fine-tuning with an anchor strategy. Experiments on pursuit–evasion and multi-agent particle environments show that MGCE produces interpretable capability analyses and achieves better performance prediction than existing baselines such as matrix factorization, IRT, and NeuralCDM.

**Strengths:**

The paper is well written and logically structured. It maintains a clear and consistent flow from problem motivation to methodology, experiments, and conclusion. Each section naturally connects to the next, making the overall argument easy to follow. The writing is fluent and professional, with sufficient background and related work to situate the contribution. Moreover, the experimental section is detailed and convincing: the authors conduct comprehensive evaluations on multiple environments, use diverse baselines (including matrix factorization, IRT, and neural cognitive diagnosis models), and report multiple quantitative metrics (MAE, RMSE, R²). These results are presented clearly through figures and tables, effectively supporting the claimed advantages of the proposed method.

**Weaknesses:**

Overall, the framework proposed in this paper appears overly complex. The authors assume that each game task $T$ must be divided into $d$ capability dimensions and $S$ scenario types. Unless the problem itself is of high importance, or the authors can automate the partitioning of capabilities and scenarios using LLMs, such a design would significantly limit the model’s generality and scalability.

In addition, I find the learning process of the “scenario–capability correspondence” described in Section 3.2 confusing. If the paper only focuses on the two environments used in the experiments, manual annotation should be sufficient. However, if the goal is to generalize to a wider range of tasks, the authors should provide the corresponding dataset and the detailed procedure of learning this mapping through the Transformer+MLP network.

More importantly, the proposed algorithm has no direct connection to game strategy evaluation. The framework is essentially an extension of the single-agent reinforcement learning paradigm and does not exhibit structural characteristics of multi-agent systems. In general, the evaluation of strategies in games or multi-agent systems should be based on concepts such as equilibrium analysis, Pareto optimality, risk–return ratio, non-transitivity of strategies, and strategy evolution. Although these ideas are mentioned in related work, the main body of the paper does not truly reflect these core principles.

Therefore, I believe the proposed framework might be more valuable if applied to complex single-agent reinforcement learning scenarios—such as autonomous driving or Atari games—where the reward structure is ambiguous or extremely sparse, rather than for game-theoretic strategy evaluation. Alternatively, I hope the authors can clarify why they place such strong emphasis on game-theoretic problems.

**Questions:**

Refer to the previous section

---

### Official Review · Reviewer_h7yP · 2025-10-31

**Soundness:** 2
**Presentation:** 2
**Contribution:** 2
**Rating:** 2
**Confidence:** 2

**Summary:**

This paper proposes a novel evaluation framework MGCE to tackle the limitations of existing game-theoretic strategy evaluation, specifically their focus on macro-level aggregate metrics that obscure multi-dimensional capability differences and their inability to handle environmental shifts. MGCE first partitions game tasks into scenarios with multi-modal large-model embeddings and models strategy-environment interactions through neural networks to quantify strategies’ multi-dimensional capabilities and scenarios’ key characteristics. It then includes an anchor strategy fine-tuning module for cross-environment performance prediction of the game strategies. Experiments on 3v1 pursuit-evasion environment and the public multi-agent particle environment (MPE) show MGCE enables interpretable strategy analysis and outperforms baselines in cross-environment prediction.

**Strengths:**

1. The paper integrates cognitive diagnosis theory into game-theoretic strategy evaluation.

2. Unlike existing evaluations that compress strategy performance into a single aggregate metric, this paper proposes the novel MGCE framework to quantify multi-dimensional capabilities and model environmental characteristics. The combination of large-model-guided multi-modal scenario embedding with neural network-based interaction modeling is an extension of existing representation learning.

3. This work performs experiments in two environments against six baselines, e.g., SVD, PMF, IRT, etc., and employs three evaluation metrics, i.e., MAE, RMSE, R2.

**Weaknesses:**

1. The experiments test two simplistic shift types: terrain shift (adding obstacles) and policy changes (modifying evader strategies), while MPE only considers terrain shift. However, they do not consider more complex shifts that align with real-world scenarios, such as dynamic rule changes (e.g., altering win conditions), and the scaling of agent numbers (e.g., from 3v1 to 5v2 in pursuit-evasion game). This limits the generalizability of MGCE’s prediction capability, as it remains unclear how the framework performs under more diverse shift types.
2. The paper defines the scenario representation rj as rj=Φ(ej), where ej denotes multi-modal features and Φ is a transformer model with average pooling. However, critical implementation details for the multi-modal fusion module Φ are omitted, which hinders the reproducibility. And it does not clarify if the transformer assigns learnable weights to each modality.
3. The paper provides minimal details on how scenarios themselves are sampled for dataset construction, creating ambiguity about whether the dataset covers representative variations of game environments. And it does not conduct any checks to quantify or mitigate potential dataset bias, which leaves uncertainty about whether results are generalizable beyond the sampled scenarios.
4.  The paper claims MGCE’s anchor strategy fine-tuning for cross-environment prediction is a key contribution, but this module is a routine application of transfer learning and few-shot adaptation. The balance score for anchor selection in this paper is a trivial heuristic for choosing a diverse anchor.
5.  The paper asserts MGCE enables interpretable analysis of strategy strengths and weaknesses. However, this "interpretability" is superficial that MGCE only labels capability dimensions but provides no evidence that these labels correspond to meaningful behavioral traits of the strategies. And the paper claims MGCE reduces human and material costs of repeated testing, but it provides no evidence of cost savings or real-world utility, e.g., it does not compare MGCE’s computational cost to baseline methods.
6.  For the MPE dataset’s terrain shift, the only shift type tested in MPE, Table 1 shows SVD outperforms MGCE in two of three core metrics. The paper fails to explain why a simple baseline outperforms MGCE in MPE.

**Questions:**

1. Please refer to the weakness section.s
2. Any ablation study on the capability association vector qj?
3. Any computational efficiency analysis for large-scale scenarios?

---

### Official Review · Reviewer_cSob · 2025-10-31

**Soundness:** 2
**Presentation:** 2
**Contribution:** 2
**Rating:** 4
**Confidence:** 4

**Summary:**

This paper introduces MGCE, a novel framework for evaluating game-theoretic strategies across multiple capability dimensions and predicting their performance under environmental shifts. Inspired by cognitive diagnosis models in educational psychology, MGCE partitions game tasks into capability-specific scenarios, embeds them via multimodal large models, and learns strategy–environment interactions through neural networks. It further employs an anchor-strategy fine-tuning mechanism to generalize performance predictions across unseen environments. Extensive experiments on both custom and public benchmarks (Pursuit and MPE) demonstrate that MGCE outperforms traditional baselines (e.g., Elo, IRT, matrix factorization) in both interpretability and prediction accuracy.

**Strengths:**

- The paper addresses a timely and underexplored problem—fine-grained, interpretable evaluation of strategic capabilities beyond win-rate aggregation.

- The integration of cognitive diagnosis with multimodal embeddings and neural interaction modeling is novel and technically sound.

- The experimental setup is comprehensive, including ablations and comparisons across diverse algorithms and environments.

-  The framework is general and potentially applicable to other multi-agent domains.

**Weaknesses:**

- The reliance on expert-defined capability-scenario mappings may limit scalability and objectivity.

- The paper lacks discussion on failure cases or limitations of the 3PL-based performance model.

- The paper lacks discussion on the differences and the relationship between the current definition,  policy evaluation protocols, and reward shaping. Policy evaluation can also measure the capabilities of different strategies, while reward shaping can take multiple dimensions into consideration, rather than focusing solely on a single win-loss rate.

**Questions:**

Apart from the weakness, I have one more question:
- The performance data used in this paper are the mean normalized time steps in two environments while the cognitive diagnosis models like IRT/DINA are basically probabilistic (modeling posterior probabilities). It does not seem very fair. Can authors provide more convincing experimental results?

---

### Official Review · Reviewer_qmQm · 2025-11-10

**Soundness:** 2
**Presentation:** 1
**Contribution:** 2
**Rating:** 2
**Confidence:** 4

**Summary:**

This paper proposes to study the evaluation of gaming strategies for their capabilities. Instead of focusing on only the win-loss performance/outcomes, the proposed method particularly aims at the evaluation across multiple dimensions of game capabilities (i.e. the strategies are evaluated in finer granularities). Moreover, it further decomposes the single environment into various scenarios in which the strategy-environment interactions can be better modeled, hence the performance of game strategies under different environment can be better predicted.

**Strengths:**

+ The proposed method shows better prediction ability upon performance, in comparison to various baselines.
+ The proposed method provides the analysis of gaming strategy on the basis of finer granularity of capabilities.

**Weaknesses:**

- The partition of game task into scenarios are based on prior knowledge and manual decision for the binary mapping between scenarios and capabilities (i.e. each scenario corresponding to the examination of a particular capability), thus being hard to generalize to other game tasks. Moreover, the feature representation for the scenarios seem to be vague, in which how to systematically construct the rules or constraints for distinctively describing each scenarios in text is unclear, and the image features extracted from different scenarios (via predefined image encoders) could also be undiscriminative (where the images for different scenarios might look quite similar to each other). Furthermore, the manual introduction of the prior knowledge (i.e. mapping between scenarios and capabilities) and the resultant richer representations of scenarios would lead to the concern of unfair comparison with respect to the baselines.
- Moreover, the description of the proposed is hard to follow: 1) both the game strategy and game scenario are modeled as one-hot vectors, but there lacks for clear explanation upon the definition of each dimension in such one-hot vectors; 2) the intuitions behind difficulty, discrimination, and chance floor as well as their necessity are not well clarified; 3) the reason why the performance can be modeled via a particular three-parameter logistic formation upon capability vector, difficulty, discrimination, and chance floor is not explained; 4) the mapping from performance back to the capabilities (e.g. for retrieving the results as in Figure 4) is not well described.
- The testing environment is built via introducing random perturbations into each scenario, such testing environment seems not well reflect the claim of "environment shift" as being not realistic enough (the testing data distribution would largely overlap with the one seen in the training time, thus no significant shift in the environment exists to support the claim).
- Although showing better quantitative results with respect to other baselines, the proposed method (i.e. MGCE and MGCE-A) still has significant variance deviated from the groundtruth (as what can be observed in the scatter plots in Figure 5), it hence is hardly to be believed for serving as a practical evaluation method.
- The model variants to conduct ablation study (with results provided in the Table 2) do not present significant differences hence the contribution of each representations (i.e. image, text, and situational representations) is not well validated.

**Questions:**

The authors should carefully address the aforementioned weaknesses (i.e. unclear description for the proposed method, the advantage brought by the manual introduction of prior knowledge which would lead to unfair comparison, the concern upon the generalizability, the skeptical claim upon handling the environment shift, and insignificant experimental results) in the rebuttal to overturn the current negative rating.

---

### Note · Authors · 2025-12-02

I have read and agree with the venue's withdrawal policy on behalf of myself and my co-authors.